# Research Advances and Application Prospect of Low-Temperature Plasma in Tumor Immunotherapy

**Xiangni Wang [1,2], Xingmin Shi [1,2,]\* and Guanjun Zhang [3,]\***

1 School of Public Health, Xi'an Jiaotong University, Xi'an 710061, China; wangxiangni0608@163.com
2 Key Laboratory for Disease Prevention and Control and Health Promotion of Shaanxi Province, Xi'an 710061, China
3 School of Electrical Engineering, Xi'an Jiaotong University, Xi'an 710049, China
\* Correspondence: shixingmin142@163.com (X.S.); gjzhang@xjtu.edu.cn (G.Z.); Tel.: +86-186-2951-4608 (X.S.); +86-158-0294-5353 (G.Z.)

**Abstract:** As an emerging technology, low-temperature plasma (LTP) is widely used in medical fields such as sterilization, wound healing, stomatology, and cancer treatment. Great achievements have been made in tumor therapy. In vitro and in vivo studies have shown that LTP has anti-tumor effects, and LTP is selective to tumor cells. Studies in recent years have found that LTP can activate dendritic cells (DC), macrophages, T cells, and other immune cells to achieve anti-tumor effects. This paper reviews the current status of tumor immunotherapy, the application of LTP in antitumor therapy, the activation of antitumor immunity by LTP, the possible mechanism of LTP in antitumor immunity, and meanwhile analyses the prospect of applying LTP in tumor immunotherapy.

**Keywords:** low-temperature plasma; tumor; immunotherapy





## 1. Introduction

Recent years have seen great breakthroughs in tumor immunotherapy, which has become an optional treatment for patients with metastatic and recurring tumors. Immunotherapy improves the body's immune response by adjusting its natural defense mechanism, thereby achieving anti-tumor effects. Low-temperature plasma is an ionized gas close to room temperature, which can generate a large amount of reactive oxygen and nitrogen species (RONS). LTP can directly fight tumors by damaging DNA, inhibiting cell proliferation, and causing cell apoptosis [1]. It can also stimulate anti-tumor immunity by regulating the functions of immune cells [2,3]. Therefore, LTP has become a new treatment of tumors. This article will review the research advances and application prospect of LTP in cancer immunotherapy.

## 2. Current Status of Cancer Immunotherapy

As a major public health hazard, cancer is a devastating disease. A variety of therapies such as surgery, radiotherapy, chemotherapy have been developed. However, the side effects of these therapies are often unavoidable and debilitating. Tumor immunotherapy has recently gained attention. Significant advances have been made in tumor immunotherapy, which has the advantages of few side-effects and solid curative effects. Immunotherapy can activate the immune response to eradicate tumor cells. It can generate systemic, specific, and long-term anticancer immunity. Vaccines, cytokines, antibodies, and immune cells are used in tumor immunotherapy to enhance the specificity and memory of the immune system against tumor cells, so as to achieve durable treatment with minimal toxicity [4]. Tumor vaccines can activate and expand tumor-specific T cells [5]. Identified key players in the anti-tumor immune response are dendritic cells (DCs), natural killer (NK) cells, and T cells. As DCs can activate T cells, DC vaccines have become one of the promising methods for tumor immunotherapy. NK cells can kill tumor cells without damaging normal tissues. The expansion and adoptive transfer of allogeneic NK cells have been

used in patients with acute myeloid leukemia and has achieved therapeutic results [6]. In adoptive T cell therapies, patients' T cells are obtained and cultured in vitro, which then reenter the patients' body to produce an immune-mediated anti-tumor response [7]. At present, many antibodies against cellular immune checkpoints (such as PD-1/PD-L1) have been developed to promote the activation of T cells and to control tumors. This treatment strategy has been proven to be particularly effective for tumors with high numbers of mutations [8]. With deepening understanding on the immune system and continuous improvement in technology, immunotherapy will play an increasingly important role in tumor treatment.

## 3. Applications of LTP in Anti-Tumor Therapy

As one of the four states of the basic matter of the universe, plasma is an ionized gas produced by the decomposition of polyatomic gas molecules or the removal of electrons from a monoatomic gas shell. It is composed of ions, electrons, atoms, ultraviolet, visible light, infrared radiation, neutral molecules, and free radicals [9]. The temperature of plasma is determined by thermal motions of electrons and heavy particles such as atoms and ions. In common thermal plasma, when the density of particles is high, due to intensive collisions among electrons and heavy particles, all particles reach thermal equilibrium. The temperature in such plasma is high, over several thousand degrees. If the atmospheric pressure plasma discharge is fast, the electrons and heavy particles are in a thermal non-equilibrium. In this case, the temperature of the particles is much lower than that of the electrons. We call such plasma low-temperature plasma [10].

Since the first report on the killing effect of LTP on melanoma in 2007 [11], the application of LTP in cancer treatment has experienced fast growth. We have previously demonstrated that in vitro LTP inhibited cell viability of HepG2 in a dose- and time-dependent manner and induced HepG2 cell autophagy [12]. In our previous study, a helium atmospheric pressure plasma jet (APPJ) was used to generate plasma-activated saline (PAS) and plasma-activated medium (PAM). PAS was injected subcutaneously to treat B16-tumor bearing mice in vivo and PAM was used to treat B16 cells in vitro. The results demonstrated that LTP induced melanoma apoptosis in vitro and in vivo [13]. Other studies have reported that local treatment of mice with LTP can shrink tumors and increase survival time [14–16]. Compared with other anti-cancer methods such as chemotherapy and radiation therapy, LTP is advantageous in its selective anti-cancer ability [17]. That is, when treating tumor cells and normal cells with the same defined dose of LTP, LTP can kill tumor cells without damaging normal ones (Figure 1). Meanwhile, LTP-based cancer therapy is unlikely to result in drug resistance, but able to accurately kill tumors deep in the body [18]. Therefore, LTP has become a new means of treating tumors.

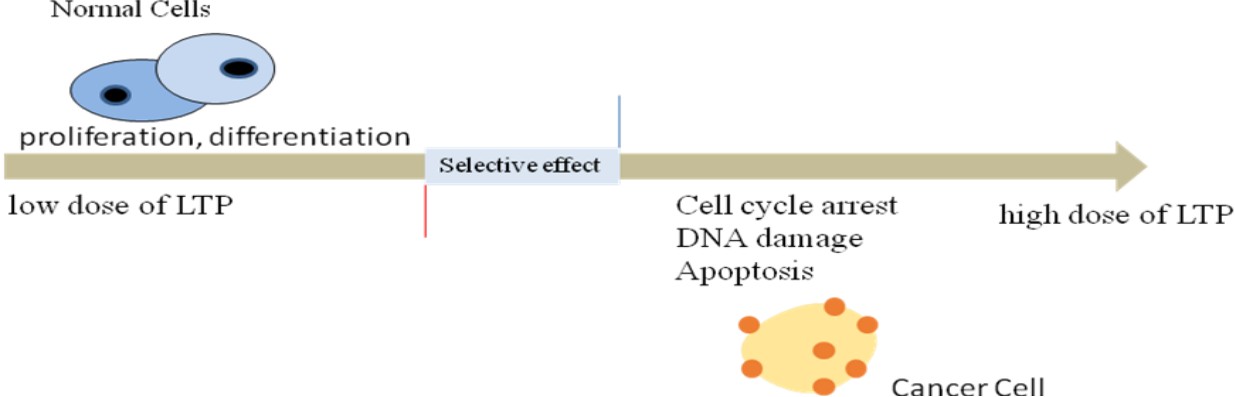

**Figure 1.** Selective effects of the optimal dose of LTP. Low doses of LTP can increase the proliferation and differentiation of normal cells, while high doses can cause cell cycle arrest, DNA damage, apoptosis to kill cancer cells.

## 4. Activation of Anti-Tumor Immunity by LTP

The development of cancer immunotherapy provides patients with better hope for treatment. Studies have shown that LTP has an effect on the activity of immune cells and immune response [19–21]. Two lines of research are currently pursued to disentangle the effects of plasma treatment in anti-cancer immunity: the ability of LTP to affect immune cells directly [22], and indirect activation of immune cells via LTP-mediated tumor cell death and pro-inflammatory signals in the microenvironment [23].

### 4.1. The Impact of LTP on Non-Specific Immune Responses

Dendritic cells, macrophages, monocytes, NK cells are involved in the non-specific immune response against tumors [24].

### 4.1.1. LTP Promotes Antigen Presentation of DCs

Dendritic cells have been recognized as the most potent antigen presenting cells, and the initiator and regulator of the body's immune response. Van Loenhout et al. [25] treated pancreatic cancer cells with LTP-activated phosphate buffered saline solution four hours and then co-cultured with the treated-cells with DC at a ratio of 1:1. They found that the maturity and antigen presentation of DCs were enhanced, but their viability was not affected. Miebach et al. [26] recently reported that colorectal cancer cells treated with an argon-based plasma jet upregulated the expressions of CD80 and CD86 of monocyte-derived DCs (moDCs) in co-culture. Tomic et al. [27] demonstrated that PAM-A375 lysate potentiated the maturation of DCs by up-regulating the expressions of CD83 and CD86. Moreover, in co-culture with allogeneic T cells, DCs loaded with PAM-lysates increased the proportion of cytotoxic T cells. Bekeschus et al. [28] found that moDCs treated with Argon plasma expressed high levels of the costimulatory molecules, including CCR7, CD25, CD40, CD86, CD83 and HLA-DR. Therefore, LTP can be used in vitro to promote the maturation of DCs, increase their number, enhance the functions of mature DCs, and activate immunotoxic T cells, so as to achieve anti-tumor effects.

### 4.1.2. LTP Enhances the Anti-Tumor Effects of Macrophages

Macrophages are one of the important components of the innate immune system, essential for the local balance of inflammation and anti-inflammation. They have anti-tumor effects [29]. Macrophages have the characteristics of high plasticity, local tissue function specificity, and abnormal differentiation induced by inflammatory factors. Tumor-associated macrophages can develop into either cytotoxic M1 or M2 macrophages depending on the tumor microenvironment. M1 macrophages are able to kill tumor cells, while M2 macrophages produce factors that suppress T cell proliferation and activity [30]. By reducing M2 and enhancing the function of M1, tumors caused by an impaired immune response can be controlled. Nagendra et al. [31] found that LTP acted as an immunomodulator for immune cell activation, stimulating the polarization of M1/M2 macrophages. The co-cultivation of tumor cells and LTP-activated macrophages reduced the invasiveness of tumor cells, and polarized macrophages weakened the maintenance ability of tumor stem cells. LTP-treated medium increased the number of macrophages and decreased that of M2. Liedtke et al. [32] injected plasma-activation solution into pancreatic mice abdominal cavities, and found that the expression of CD206 (M2 marker) was significantly reduced. Miller et al. [33] found that low-plasma-treated macrophages showed an increased migratory activity and mediated tumor cell killing in a TNF-$\alpha$-mediated manner. Khabipov et al. [34] studied plasma exposure of pancreatic ductal adenocarcinoma (PDA), and subsequent co-culture with macrophages significantly reduced the number of macrophage clusters compared to untreated PDA cells. Therefore, tumor therapy by targeting pro-inflammatory macrophages may become an attractive anti-tumor treatment strategy.

### 4.1.3. LTP Effects Other Non-Specific Immune Cells

Neutrophils are one of the immune cells existing in the tumor microenvironment. Tumor-associated neutrophils have been proven to have functional plasticity, whose functions and phenotypes are polarized differently at different stages of the tumor [35]. The formation of neutrophil extracellular traps (NETs) increased after LTP treatment [36]. Xu Dehui et al. [37] used LTP-activated water to gavage immunodeficient mice, and found that neutrophils and monocytes in the blood slightly increased. Liedtke et al. [32] injected PAM into mice seven days after tumor challenge and analyzed the changes in intratumoral immunological profile. The results demonstrated the number of neutrophils increased in the treatment group. However, there are no reports on the effects of LTP on other innate immune cells such as NK cells and mast cells. Our team has explored the effects of plasma-activated saline on NK cells and found that an appropriate dose of LTP-activated saline promoted the vitality and killing activity of NK-92MI cells, while excessive doses inhibited their viability and killing activity.

### 4.2. The Impacts of LTP on Specific Immune Responses

T cells and B cells play an important role in the specific immune response. Freund et al. [38] found that LTP-treated saline increased the activity of T cells. Long-time LTP treatment caused the death of many cells, whereas T cells still maintained the ability to activate and divide. Haertel et al. [19] analyzed the effect of dielectric barrier discharge (DBD) on subsets of lymphocytes and found that after a short time (5 s) of LTP exposure, the number of T cells increased slightly, indicating that short-term LTP exposure promoted the proliferation of T cells. However, with an increase in treatment time, the number of T cells began to decrease after 20 s, while the number of Th cells started to decrease after 60 s. The increase of B cells indicated that the lymphocytes were selectively sensitive to LTP depending on the time of LTP treatment. Liedtke et al. [32] identified a significant increase of T cells in murine tumors repeatedly exposed to plasma-treated medium.

### 4.3. The Effects of LTP on Other Immune Cells

In addition to effecting the function of non-specific and specific immune cells, LTP can also promote activity of mixed cells extracted from immune organs and enhance anti-tumor effects. A study reported that LTP-treated prostate cancer cells increased the viability and toxicity of bone marrow cells [39]. Rödder et al. [40] found when LTP-treated mouse spleen cells were co-cultured with B16-F10 cells, spleen cells produced more inflammatory cytokines and enhanced the anti-tumor effects.

## 5. Anti-Tumor Immune Mechanism of LTP

### 5.1. The Oxidative Stress Toxicity of LTP

LTP produces oxidative stress. RONS, charged particles, ultraviolet radiation and electromagnetic fields, all of which can play a role and produce synergy in the therapy of cancer [41]. Among them, RONS is the most important element of LTP to kill tumors. LTP treatment increased intracellular ROS [42], which causes DNA double strands to break [43]. DNA damage results in cell apoptosis [44] and decreases the cell viability of tumor [45] (Figure 2).

### 5.2. The Relationship between RONS and Immune Response

Studies have shown that RONS participated in immune response. For example, $H_2O_2$, $O_2^-$, OH, NO, $ONOO^-$ and $NO_2^-$ participated in the phagocytosis of microorganisms of macrophages. NO affected the synthesis and secretion of cytokines (TNF-$\alpha$, IFN-$\gamma$, IL-1, IL-6, etc.) of immune cells (macrophages, NK cells, T cells). RONS increased the number of macrophages [46]. Some RONS affected the cross-presentation antigen of dendritic cells, migration of macrophages, and regulation of cytokine receptor expression [7]. Mizuno et al. [47] treated B16-F10 melanoma with plasma. The tumor grew on both legs, but only the tumor on the right side was directly exposed to the plasma. It found that

the tumor growth on both sides was inhibited. This result indicated that plasma could trigger the innate immune response through signal transmission. Therefore, we infer that RONS produced by LTP would cause changes in the immune response via related signaling pathways.

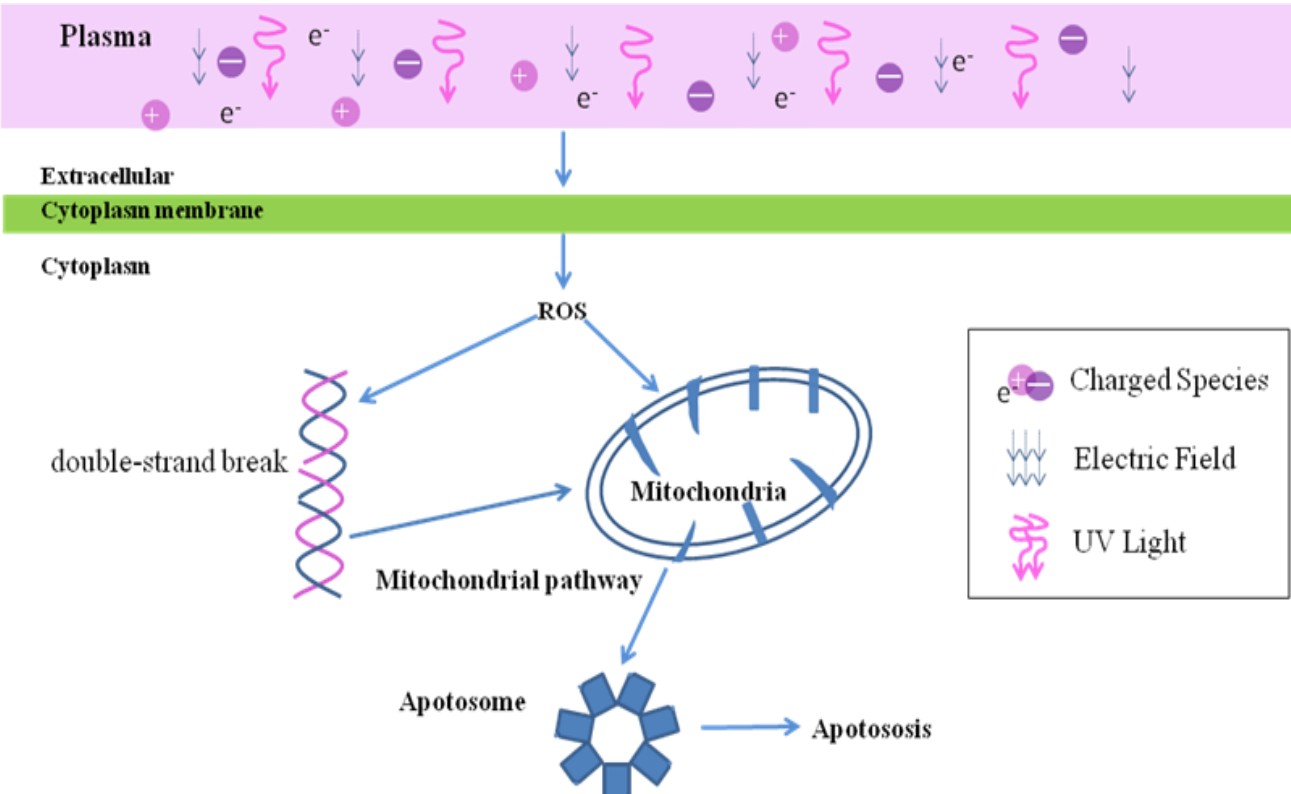

**Figure 2.** The oxidative stress toxicity of LTP. LTP cause the increase of intracellular ROS, which leads to DNA damage and mitochondrial damage, thus leading to the cell apoptosis.

### 5.3. LTP Induces Immunogenic Cell Death

Immune tolerance refers to the fact that tumor cells lack one or more components which are necessary to effectively stimulate the body's immune system. When some physical and chemical factors induce the apoptosis of tumor cells, tumor cells are transformed from non-immunogenic to immunogenic, which is called immunogenic cell death (ICD). ICD activates specific T lymphocytes and specific immune response to eliminate tumors. When ICD occurs, immunostimulatory molecules damage associated molecular patterns (DAMPs), are released from or displayed by the dying cells. Calreticulin (CRT), Adenosine Triphosphate (ATP) and High Mobility Group Box 1 (HMGB1) are well-known DAMPs that are released outside cells in response to ICD. More and more studies have shown that the oxidation produced by LTP induces immunogenic death of cells, causing an increase of DAMPs. In fact, ATP serves as a "Find Me" signal and CRT acts as an "Eat Me" signal for immune cells [48]. The DAMPs stimulate an immune response by stimulating antigen-presenting cells. The uptake of tumor antigens by dendritic cells was increased. Then, DCs presented tumor antigens to anti-tumor T cells [41] (Figure 3). Lin [49] et al. used non-thermal plasma to induce ICD in A549 lung cancer cells, and found secreted danger signals from cells undergoing immunogenic death enhanced the anti-tumor activity of macrophages.

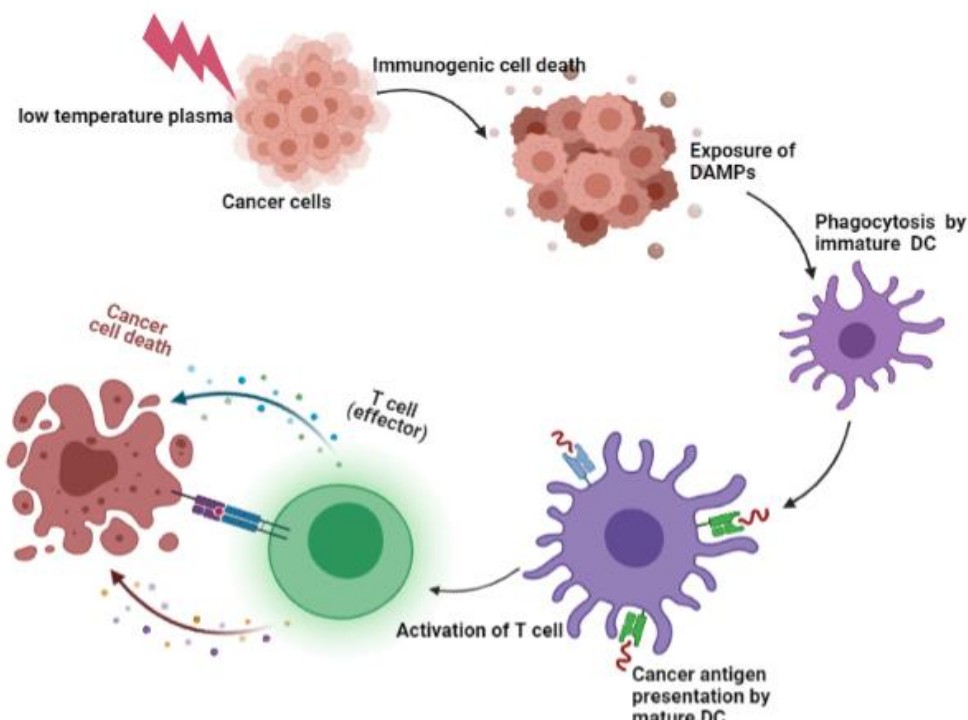

**Figure 3.** Immunogenic cell death and the activation of T cell. LTP induced immunogenic death of cells, causing an increase of DAMPs. The DAMPs activate dendritic cells. Then, mature dendritic cells presented tumor antigens to anti-tumor T cells.

### *5.4. LTP Regulates the Tumor Microenvironment*

Immune escape is one of the important mechanisms of malignant tumors. The tumor microenvironment (TME) is considered the main reason for immune suppression and evasion of immune surveillance. The tumor microenvironment plays an important role in the survival, growth, invasion, and metastasis of tumor cells. The immunosuppressive molecules and inhibitory molecules in the TME can affect the function of immune cells [50]. Due to the tumor microenvironment, many targeted therapies have failed to achieve desired results. LTP not only affects the tumor cells themselves, but also regulates the TME. It has been observed that long-term LTP treatment inhibited cell viability and collagen production of murine fibroblasts [51]. LTP could lead to increased cytotoxicity and a sharp decrease in the release of VEGF. Nagendra et al. [31] found plasma exposure not only decreased M2 macrophages, but also remarkably increased the expressions of IL-1α, IL-1β, IL-6 and TNF-α. The microenvironment is changed by LTP to prevent the growth and metastasis of tumors, and to inhibit immune cells, achieving a lasting therapeutic effect.

### 6. Prospects and Challenges

It has been proven that LTP can treat a variety of malignant cancers. LTP can kill tumors directly, and can kill cancer cells by affecting the functions of immune cells such as DC, macrophages and T cells. LTP can induce immunogenic cell death of tumor cells, regulate tumor microenvironment, and increase immune response to kill tumors. Therefore, LTP stimulating the immune response is key to improving the cure rate of patients. However, current research on LTP in cancer immunotherapy has the following problems. First, studies of LTP in immunotherapy are still in their infancy, most of which are in vitro tests. Further animal or clinical experiments are needed to explore the anti-tumor effects of LTP by activating immune cells. Second, the underlying mechanism of LTP stimulating the immune response is still unclear. Third, as immune cells are affected by LTP in a time-/dose-dependent manner, it is difficult to determine the optimal LTP dose to activate the body's immune response. Our team [19] treated peripheral blood lymphocyte samples of 20 healthy adult volunteers with LTP, and found that when low doses of LTP were used,

lymphocyte activity was activated. Fourth, immune cells have different sensitivities to LTP. Therefore, when LTP is used to stimulate the immune response, the key is to use the best time and method to treat different immune cells. This is also a huge challenge for LTP in tumor immunotherapy.

**Author Contributions:** Conceptualization, X.S. and G.Z.; writing—original draft preparation, X.W.; writing—reviewing and editing, X.S. and G.Z. All authors have read and agreed to the published version of the manuscript.

**Funding:** This research was funded by the General Program of National Natural Science under Grant NO. 51677146 and the Shaanxi Province Key Research and Development Program (Grant NO. 2018ZDCXL- SF-02-03-01).

**Institutional Review Board Statement:** Not applicable.

**Informed Consent Statement:** Not applicable.

**Data Availability Statement:** Data sharing not applicable.

**Acknowledgments:** We would like to thank Feng-Ling Peng for providing linguistic assistance for this manuscript.

**Conflicts of Interest:** The authors declare no conflict of interest.

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
