# Peer review of "Research Advances and Application Prospect of Low-Temperature Plasma in Tumor Immunotherapy"

_applsci, doi:10.3390/app11209618_

Round 1
Reviewer 1 Report
The authors reviewed the the low-temperature plasma application in tumor immunotherapy, which has been a hot topic in the field recently.
The review is brief and to the point. The logic and organization of the review is sound and clear. However, the authors need to improve in the following major aspects:
- English is OK but not professional. Many sentences are understandable but need to be rephrased and polished.
- Most of the references are more than 5 years old. As a review article, it is important to review the state-of-art of the topic. For example,
- in Section 2. Current status of Cancer Immunotherapy, references are ranged from 2005-2017, it is hardly "current".
- The same problem goes for Section 3, application of LTP in anti-tumor therapy. Most references of 11-17 are old articles. It would add great value to the article if the authors can review the current development of LTP in anti-tumor therapy.
Author Response
Dear Editors and Reviewers,
Thank you so much for your comments and suggestions. In accordance with your suggestions, we have revised our manuscript carefully. The responses to reviewer 1’s comments are listed in the attachment. Please see the attachment.

Reviewer 2 Report
- There is no space through the whole manuscript before all reference number.
- Line 67: Please include the reference about melanoma published in 2007 into reference lists.
- Line 71: please add after “in vitro and in vivo” experimental models.
- Figure 1. Under selective effect should be added “optimal” or “ medium” dose of LTP, and please include in the text Line 74: “ with the same optimized/defined/medium dose of LTP…”
- Reference 17 is not about drug resistance and LTP based therapy.
- Line 100: explain the PAM abbreviation.
- Line 99 and 101: please correct “increased the expression level of CD80 and CD86”
- Line 103: mo DC – explain mo abbreviation
- Line 104: please correct “increase of costimulatory molecules expressions level”
- Line 113: please correct “can differentiate into cytotoxic M1 type macrophage or M2 type macrophage which…”
- Line 122: add Miller et al [32] like it was written before.
- Line 124: explain PDA abbreviation.
- Line 134: “Liedtke et al [31] found an in tendency increased number of neutrophils in tumors of the plasma group”, correct the sentence
- Line 144: DBD abbreviation explain,
- Line 147: To clarify the process maybe instead of saying “But with the increase of treatment time, the number of T cells began to decrease (20 s), T cells and Th cells decreased after 60 s” should be “ But with the increase of treatment time, the number of T cells began to decrease after 20 s, while number of Th cells started decreased after 60 s”?
- Line 214. Psoriasis is an autoimmune disease - not the tumor model, therefore the last reference should be removed and some references about tumor microenvironment and LTP should be added in this section.
Author Response
Dear Editors and Reviewers,
Thank you for reviewing our manuscript and offering valuable advice. In accordance with your suggestions, we have made the following revisions to our manuscript. The responses to reviewer 2’s comments are listed in the attachment. Please see the attachment.

Round 2
Reviewer 1 Report
The manuscript has been revised according to the reviewer's suggestions.